https://doi.org/10.1038/s41467-020-15657-3　　**OPEN**

# Genomic release-recapture experiment in the wild reveals within-generation polygenic selection in stickleback fish

Telma G. Laurentino[1]✉, Dario Moser[1], Marius Roesti [2], Matthias Ammann[1], Anja Frey[1], Fabrizia Ronco [1], Benjamin Kueng[1] & Daniel Berner[1]✉

How rapidly natural selection sorts genome-wide standing genetic variation during adaptation remains largely unstudied experimentally. Here, we present a genomic release-recapture experiment using paired threespine stickleback fish populations adapted to selectively different lake and stream habitats. First, we use pooled whole-genome sequence data from the original populations to identify hundreds of candidate genome regions likely under divergent selection between these habitats. Next, we generate F2 hybrids from the same lake-stream population pair in the laboratory and release thousands of juveniles into a natural stream habitat. Comparing the individuals surviving one year of stream selection to a reference sample of F2 hybrids allows us to detect frequency shifts across the candidate regions toward the genetic variants typical of the stream population—an experimental outcome consistent with polygenic directional selection. Our study reveals that adaptation in nature can be detected as a genome-wide signal over just a single generation.

[1] Department of Environmental Sciences, Zoology, University of Basel, Vesalgasse 1, 4051 Basel, Switzerland. [2] Institute of Ecology and Evolution, University of Bern, Bern, Switzerland. ✉email: telma.laurentino@unibas.ch; daniel.berner@unibas.ch

Adaptation to novel environments can occur rapidly[1–4], with evolution in ecologically relevant phenotypes arising within a few generations[5–8]. Such rapid phenotypic evolution has sometimes been linked to changes in allele frequencies at underlying genetic loci[9–12], although this generally concerns a small number of loci harboring genetic variants of large phenotypic effect[13]. However, adaptation commonly involves a great number of loci spread across the genome[14,15], and how rapidly natural selection influences ecologically important loci genome-wide remains largely unexplored empirically outside prokaryotic organisms[16–18]. To address this gap, we here investigate a rapid genome-wide response to selection under natural experimental conditions in threespine stickleback fish (*Gasterosteus aculeatus*).

In this organism, the postglacial colonization of freshwater by marine ancestors has led to the evolution of distinct ecotypes residing in adjacent, selectively different lake and stream habitats[19–23]. Such divergent lake-stream adaptation has occurred in stickleback within the Lake Constance basin in Central Europe[22,24–26]. In this system, the ecotype inhabiting Lake Constance exploits the pelagic (open-water) foraging niche, whereas multiple tributary streams harbor ecotypes with a benthic (bottom-feeding) lifestyle[22,27]. This ecological diversification is mirrored by divergence between the lake and stream ecotypes in traits, such as foraging and predator defense morphology, and life history[22,24,27–29].

The lake and stream ecotypes within the Lake Constance basin are undoubtedly products of adaptive evolution: transplant experiments in natural streams have revealed that stream individuals consistently outperform lake individuals (and F1 lake-stream hybrids) within a single generation, and that this fitness difference has a strong genetic basis[30]. At the molecular level, marker-based genomic investigations of natural populations from the Lake Constance basin have found signatures of divergent selection[25,26], for instance in the form of exceptionally strong lake-stream difference in the frequency of genetic variants in some genome regions, and indicated that this selection is highly polygenic (that is, involves a great number of genetic loci across the genome).

What is now needed to understand the mode and speed of adaptation at the genomic level is a manipulative experiment connecting rapid ecological adaptation to genome-wide changes in the frequency of genetic variants. We performed such an experiment in nature, involving (i) identifying genome-wide candidate target loci for divergent lake-stream adaptation using whole-genome sequencing in a natural lake-stream population pair; (ii) exposing a laboratory-bred, genetically mixed F2 hybrid population derived from this lake-stream pair to a natural stream habitat for one year; and (iii) assessing variant frequency shifts at the target loci in the survivors. Finding genome-wide evidence of directional polygenic selection in our field experiment, we finally use individual-based simulations to explore the underlying selection.

## Results

**Lake-stream stickleback under polygenic divergent selection.** A key assumption underlying our study was that if natural selection drives allele frequency shifts in an experimental population within a single generation, these shifts are likely subtle and hence difficult to detect by just comparing the experimental population before and after selection. Our strategy was therefore to define genomic regions likely to be targeted by selection during the experiment a priori. To discover such regions, we focused on a single lake-stream stickleback pair[22,24,25,30] residing within the Lake Constance basin (Fig. 1). From each population, we collected a large sample of individuals ($N = 240$ and $229$) in the wild.

These natural population samples were then subjected to pooled whole-genome sequencing at high read depth (210×), and the sequences were aligned to the 447 megabase (Mb) threespine stickleback genome and screened for single-nucleotide polymorphisms (SNPs). For each of the 977,723 autosomal SNPs discovered, we then quantified the magnitude of differentiation between the lake and stream population by the absolute allele frequency difference (AFD)[31].

This revealed a modest magnitude of differentiation between the natural populations (median AFD = 0.139, mean = 0.165). Numerous genomic regions, however, stood out clearly from this background level of differentiation, reaching maximal values up to 0.934 (Fig. 2; Supplementary Fig. 1). Nevertheless, no single SNP with fixed differences between the habitats was observed, which may reflect dispersal and gene flow between lake and stream stickleback within the Lake Constance basin[25,26], or that adaptation does not require the complete fixation of locally favorable alleles[14,32,33]. Patterns of differentiation along chromosomes were qualitatively similar to those recovered in a previous lower-resolution genome scan for the same population pair based on reduced-representation (RAD) sequencing (compare Supplementary Fig. 1 to the 'Lake vs. NID' panel in Supplementary Fig. 7 from ref. [25]). For instance, the SNP with the highest differentiation in the latter analysis (Fig. 4a in ref. [25]) also showed extreme differentiation in the present lake-stream comparison (AFD = 0.6), and an inversion on chromosome 1 emerged as highly differentiated in both studies (Supplementary Fig. 1; Fig. 6b in ref. [25]). Our comparison of the natural populations performed with whole-genome resolution clearly confirms the view that adaptive divergence between lake and stream stickleback involves differentiation in hundreds of genomic regions[25], and hence qualifies as polygenic.

From the most strongly differentiated of these regions—considered most likely to respond to selection during the release–recapture experiment, we then selected a single representative SNP (Fig. 2). These 126 total target SNPs displayed AFD values ranging from 0.477 to 0.934 (median = 0.589, mean = 0.602).

**Predicted targets of selection evolve in a single generation.** To obtain an experimental population for studying selection in action, we derived a large F2 hybrid population from our focal natural lake and stream stickleback population pair in the laboratory. Owing to random assortment and recombination, these F2 hybrids represented a genomic mixture of lake and stream ancestry (Fig. 1). From the F2 hybrid population, 3000 juvenile individuals were released into the wild at a natural stream site suitable to, but not currently inhabited by, stickleback (Fig. 1). At the same time, we took a reference sample of 510 individuals from the laboratory hybrid population to obtain a baseline of the genomic composition of the F2 hybrid population before the release. One year after the release, the F2 hybrids were recaptured in the field, recovering 37 total fish hereafter called survivors. To study evolution during the exposure to natural field conditions, both the reference sample and the survivors were subjected to whole-genome sequencing at high read depth (127 and 115×). After stringent quality filtering, these data allowed us to assess through a resampling approach whether the target SNPs predicted to be under divergent lake-stream selection showed elevated allele frequency shifts from the reference sample to the survivors compared to genome-wide neutral SNPs.

Our sequence data showed that in the reference sample characterizing the F2 hybrid population before the field release, the frequency of the stream allele (i.e., the allele showing a higher

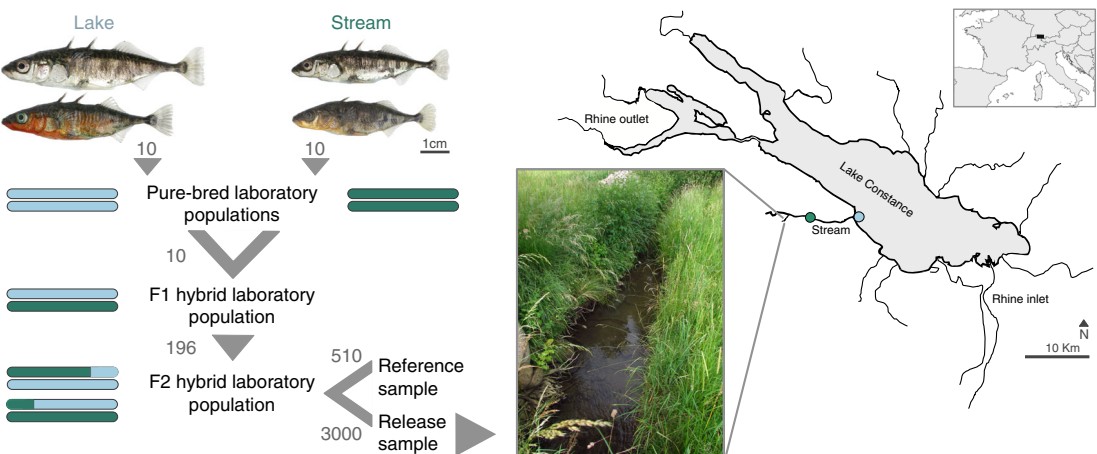

**Fig. 1 Design and sampling locations for the release–recapture experiment.** The study focuses on a natural lake and stream population pair sampled at the sites indicated by the blue (lake) and green (stream) dots in the map of the Lake Constance basin, Central Europe (map created by the authors based on data from Google Earth, Landsat/Copernicus). Representative females and males from these populations are depicted at the same scale in the top and bottom row (Photo credit: D.B.). The breeding scheme describes how F2 hybrids were derived from the pure-bred populations under laboratory conditions, with numbers specifying how many replicate families founded the subsequent laboratory population. The F2 hybrids represented a mix of lake and stream genomic ancestry, as visualized by the color-coded bars symbolizing diploid genotypes for a single chromosome. For the field experiment, a sample of 3000 juvenile F2 hybrid individuals was released into a natural stream (photo from summer 2017, credit: T.G.L.), and the survivors recaptured one year later. An additional sample of 510 individuals served as a reference to quantify allele frequencies at the time point of the release.

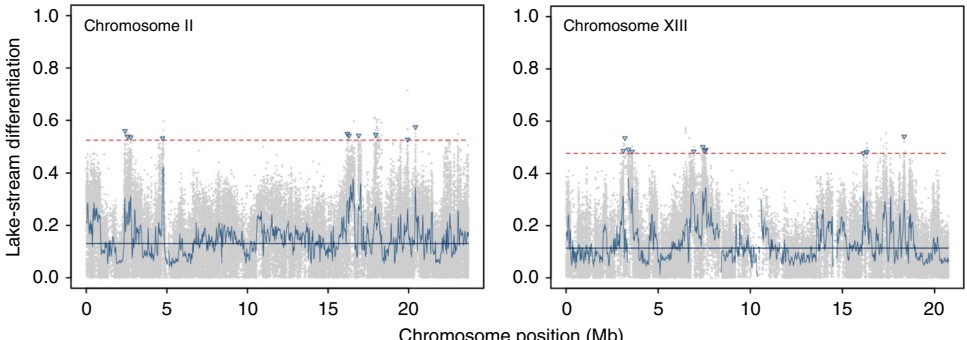

**Fig. 2 Genomic differentiation between the natural lake and stream populations.** Profiles of the absolute allele frequency difference (AFD) along two exemplary chromosomes, with the gray dots representing individual single-nucleotide polymorphisms (SNPs) and the solid black lines indicating chromosome-specific median values. The blue curves show differentiation smoothed by averaging AFD across SNPs for 40 kb sliding windows with 20 kb overlap. The dashed red lines give the upper 0.1 percentile of the chromosome-specific AFD distribution used as a threshold for delimiting candidate genome regions under divergent lake-stream selection. The triangles identify a subset of the 126 total genome-wide SNPs eventually used as experimental target SNPs representing the independent candidate regions. Note that some high-differentiation regions are not tagged by a target SNP, or the target SNP is not the one with the highest AFD value, because SNPs in these regions failed to satisfy quality filters in the reference-survivors comparison.

relative frequency in the natural stream than the lake sample) at the 126 target SNPs was almost perfectly intermediate between the frequencies of the natural lake and stream population samples (Fig. 3a). Our laboratory breeding protocol thus mixed lake and stream genomes in the F2 hybrids reliably. Note, however, that our target SNPs were generally relatively far from the fixation for alternative alleles in the lake and stream populations, and that the F2 hybrids were derived from ten independent F1 hybrid families. Hence, most of the haplotype-level diversity exposed to selection in the F2 hybrids was not generated by recombination when intercrossing the F1 hybrid generation, but pre-existed in the natural populations. During the experimental period, the majority of the target SNPs (77 out of 126; 61%) exhibited an allele frequency shift in favor of the stream allele (Fig. 3a, b), a numerical imbalance unlikely to arise by chance (two-tailed binomial probability: 0.016). The median allele frequency shift across the target SNPs was 2.5% (mean 2.3%). Resampling 126

genome-wide neutral SNPs at random 9999 times and re-calculating the median shift for each iteration indicated that observing an overall shift of 2.5% or greater in any direction was unlikely (two-tailed probability: 0.0173; based on the mean: 0.006) (Fig. 4). All these findings remained robust to changing analytical detail (robustness checks described in the Methods and summarized in Supplementary Fig. 2).

In genome regions inferred to be under divergent lake-stream selection based on the natural population samples, our genetically mixed lake-stream F2 hybrid fish exposed to natural stream conditions thus exhibited exceptionally large allele frequency shifts in the expected direction. This pattern is consistent with a slight response to polygenic directional selection within a single generation. Conversely, our experiment also confirms that the regions of high differentiation between the natural lake and stream populations detected in our genome scan are indeed under divergent selection.

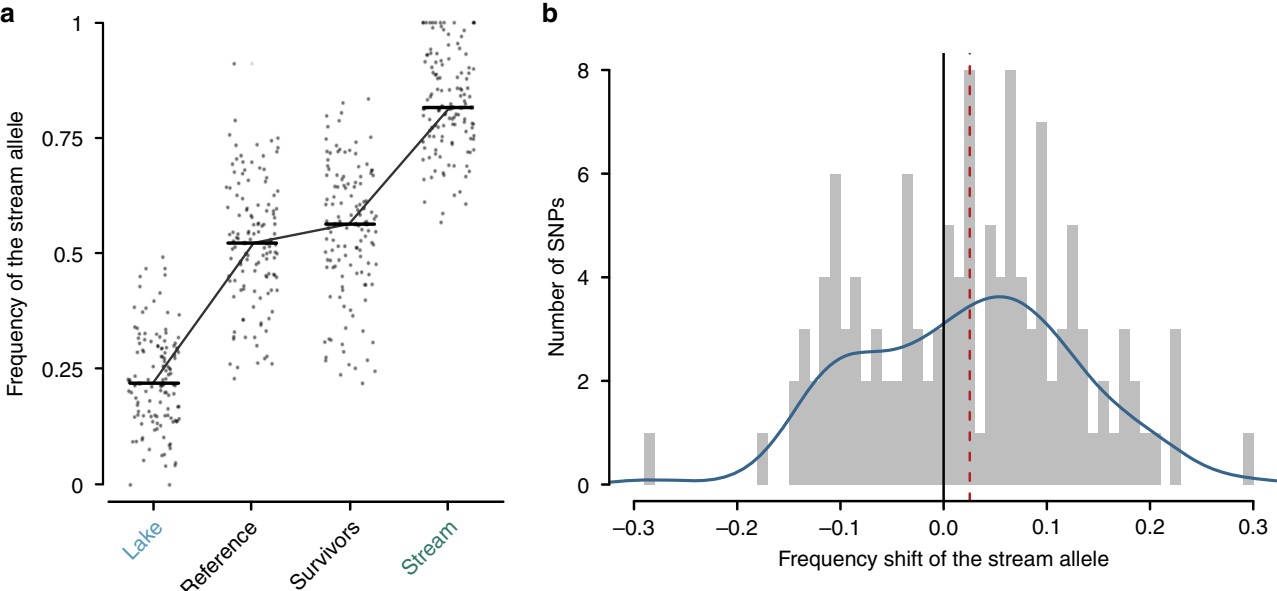

**Fig. 3 Allele frequency shifts at the target SNPs within a single generation. a** Frequency of the stream allele in the natural populations (lake and stream), in the F2 hybrid reference sample, and in the pool of the F2 hybrids surviving one year in the stream (survivors). Each point represents one of the 126 total genome-wide target SNPs, and the horizontal bars indicate the median allele frequency in each sample (lake and stream: 0.219, 0.816; reference: 0.523; survivors: 0.563). **b** Distribution of the frequency shift of the stream allele at the target SNPs during the experiment, revealing a trend toward positive shifts (i.e., the stream allele increased in frequency). The median shift of 2.5% across all target SNPs is shown by the dashed red line. The blue curve visualizes the distribution kernel density-smoothed (bandwidth: 0.04).

**Insights from simulated selection**. To develop a sense for the strength of selection at individual loci required to produce a frequency shift in the order of the one observed experimentally across the target SNPs, we tailored individual-based simulations of polygenic directional viability selection over one generation to our experiment. Because the link between multilocus genotype and fitness is unknown, we considered both multiplicative and additive contributions of individual loci to overall fitness.

These simulations revealed that with multiplicative fitness, median allele frequency shifts of the magnitude observed became likely with per-locus selection coefficients above ~0.06 (Fig. 5a). With additive fitness, however, weaker selection in the order of 0.01 already sufficed to produce frequency shifts compatible with our empirical observation (with 200 simulated loci, even weaker selection was sufficient; Supplementary Fig. 3b). The latter coincided with the domain in which we observed directional selection to become truncational (at $s = 0.009$). Strong truncation selection associated with even higher selection coefficients under additive fitness produced frequency shifts well beyond the one observed experimentally. Overall, our simulations of selection over a single generation indicate that the allele frequency shifts observed in the field experiment are plausible under directional selection on many loci; at least under some form of additive fitness, the required per-locus selection coefficients may be relatively low.

We next extended our simulations to multiple generations to explore how rapidly locally favored alleles increase in frequency when a genetically variable population is exposed to polygenic directional selection in a novel environment. This indicated that irrespective of the fitness scheme (multiplicative, additive), evolution across the selected loci was initially very rapid (Fig. 5b); allele frequency changes well beyond the empirically observed baseline lake-stream differentiation across all genome-wide SNPs (AFD = 0.14) were achieved within a few dozen generations. Assuming analogous evolution in a hypothetical population exposed to selection in the opposite direction (i.e., lake habitat),

population differentiation at the selected loci would reach 0.6— the median AFD observed empirically between the natural lake and stream population sample across the 126 target SNPs— within a few dozen generations too.

The simulations of evolution over multiple generations made several assumptions that may or may not be satisfied in a natural context (e.g., the distribution of the initial frequencies of the locally favored alleles, or that selection coefficients remain constant during the course of evolution). Nevertheless, the speed of evolution observed in these simulations is fully compatible with the rapid phenotypic evolution (and frequency changes at the few known underlying loci) observed in stickleback in the wild[4,5,34,35], thus adding plausibility to the results from the single-generation simulations.

## Discussion

Allele frequency changes observed in our stickleback release–recapture experiment suggest a polygenic response to directional selection within a single generation. Strong independent support for this interpretation derives from a previous experiment transplanting juvenile Lake Constance and tributary stream stickleback and their F1 hybrids into multiple, ecologically different streams, consistently and unambiguously demonstrating directional viability selection within a single generation[30]. The experimental fish in that study were derived from laboratory lines; hence ecotype-dependent survival was largely genetically determined. In this light, there can be little doubt that the survivors recaptured in the present experiment represent a genetically non-random subset of the F2 hybrid population initially released. Also, selective shifts in the order of magnitude observed are not only plausible, but actually required to explain allele frequency shifts at candidate adaptation loci arising during phenotypic evolution from standing genetic variation over dozens of generations in wild stickleback

exposed to novel habitats[4,5,34,35]. Indeed, our simulations of polygenic selection over multiple generations confirm that selection coefficients appearing plausible in our single-generation experiment are compatible with the rapid allele frequency shifts observed in naturally evolving stickleback. Our genomic experiment thus suggests that adaptive allele frequency shifts can be detected over a single generation when focusing on a collective signal across many loci predicted a priori to be targets of natural selection.

Given the absence of downstream dispersal barriers in our experimental stream, a possibility worth considering is that the observed allele frequency shifts may to some extent reflect

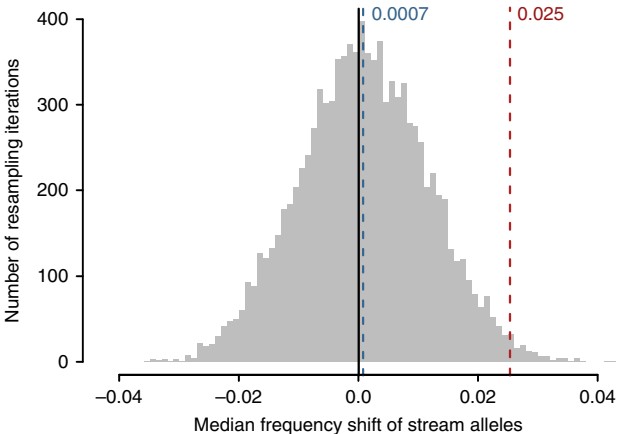

**Fig. 4 Observed allele frequency shift at the target SNPs compared to neutral SNPs.** Distribution of the median reference-survivors frequency shift of the stream allele across 126 neutral SNPs, based on 9999 replicate resampling iterations. The neutral SNPs were required to lie within an AFD range of 0–0.1 in the comparison of the lake and stream population, and were further constrained to a minor allele frequency (MAF) spectrum matching the one observed across the target SNPs by applying a MAF threshold of 0.35 in the reference sample. The dashed blue line indicates the grand median random shift. The dashed red line gives the median reference-survivors shift of the stream allele observed across the 126 target SNPs during the field experiment.

genotype-dependent dispersal. That is, experimental individuals in poor phenotypic condition due to relatively unfavorable combinations of alleles across the ecologically relevant loci may have dispersed, thus altering the genomic composition of the remaining population[36–40]. This mechanism represents a (particularly effective) form of, rather than an alternative to, natural selection, because genetically based fitness differences among genotypes are a prerequisite; habitat preference mechanisms unrelated to individual fitness, whether learned or genetically determined, are not expected to systematically shift allele frequencies in genetically mixed F2 hybrids generated under laboratory conditions. Phenotype-related habitat preference has indeed been suggested in lake-stream stickleback[41], although further evidence, including on a potential genetic basis, is needed. Such information would help understand whether genome-wide responses to selection are facilitated by dispersal behavior.

A further insight, emerging from our simulations, is that a substantial within-generation polygenic response to directional selection may be plausible despite weak selection at the level of individual loci. This holds in particular when assuming that the loci affect fitness additively, as suggested by another stickleback experiment[42]. In this case, the fate of a given allele is highly contingent on the allelic makeup at other loci within an individual. Specifically, selection can here become very effective when unfavorable alleles across all loci together drive the whole population toward an absolute mean fitness near zero. In this domain, many individuals actually do have zero fitness and selection is truncational. Individuals by chance carrying particularly favorable multilocus genotypes will then display an exceptionally high fitness relative to the population mean. Our simulation finding of a substantial response to selection despite weak per-locus selection under additive fitness supports the notion that polygenic truncation selection—including departures from strict truncation in which individuals are simply ranked by multilocus genotype, allows for strong responses to selection by eliminating unfavorable alleles jointly[43,44]. While we believe that truncation selection is plausible in our stickleback system, because of very high juvenile mortality measured under natural conditions[30], we emphasize the urgent need for more refined experimental information on the connection between multilocus genotype and fitness in this and other organisms. As long as this

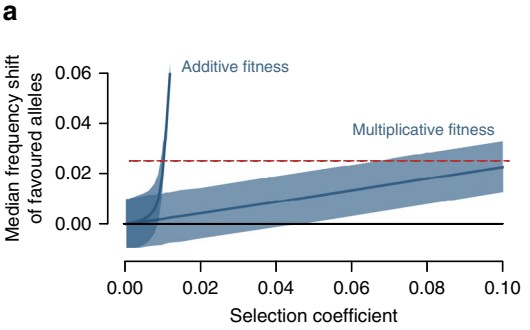
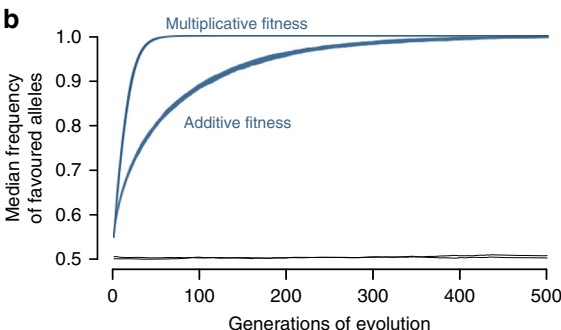

**Fig. 5 Allele frequency shifts in simulations of polygenic directional selection. a** Response to viability selection on 100 loci over a single generation for a range of per-locus selection coefficients and two different fitness schemes. For each of the 10,000 replicate simulations, the median shift of the selectively favored allele across the loci was recorded. The blue curves represent the median of these values across the replicates, and the blue bands give the associated 95 percentiles. The dashed red line indicates the median frequency shift of the stream alleles observed across the 126 target SNPs in the field experiment. **b** Response to selection on 100 loci over multiple generations, for the same two fitness schemes as in **a**. For each fitness scheme, a single selection coefficient compatible with the experimentally observed median allele frequency shift was chosen (see **a**; 0.1 and 0.01 for multiplicative and additive fitness). The blue bands display the full range across the 40 replicate simulations of the median frequency of the favorable allele across all selected loci. The black lines indicate the grand median allele frequency of the focal allele across 20 neutral loci and replicate simulations (the lines are not labeled because they largely overlap between the two fitness schemes). In both **a** and **b**, the initial frequencies of the favorable alleles at the selected loci were drawn at random from the distribution observed empirically at the target SNPs.

relationship is not better understood, it remains possible that estimates of the strength of selection at single ecologically important loci based on allele frequency changes observed over generations are inflated when numerous loci across the genome are under selection simultaneously.

## Methods

**Study system and experimental field site.** Our investigation focuses on a single lake-stream stickleback pair (the ROM lake and the NID stream populations[22,24,25,30]) residing within the Lake Constance basin (Fig. 1). For our field experiment, we required a natural stream site suitable to, but not currently inhabited by, stickleback. Such a site was identified in the headwater of the stream inhabited by the NID population. Our experimental stream was formerly piped, but opened and restored one year before the experiment. To increase water volume of this small stream, and thus carrying capacity, we constructed two successive shallow dams (Supplementary Fig. 4c). These hindered upstream dispersal and produced a total stream section of c. 50 m suitable to stickleback. Rapids downstream of the experimental site made the natural colonization of this headwater stream by stickleback highly unlikely. Accordingly, extensive minnow trapping before dam construction (April 2015) and immediately before the experimental release (September 2015) recovered no stickleback, although two other fish species (known to be efficient colonizers; *Phoxinus phoxinus* and *Barbatula barbatula*) were present. At the time of the release, the experimental stream exhibited the vegetation and invertebrate fauna typical of other stream sites within the study region, and natural predators were observed during later inspections (e.g., dragonflies and gray heron, *Ardea cinerea*; photographs of the site and further detail are presented in Supplementary Fig. 4).

**Experimental populations and release–recapture protocol.** For the identification of regions putatively under divergent selection, a sample of individuals from the lake and the stream population was needed. These samples, hereafter referred to as natural population samples, included 240 individuals from the lake and 229 individuals from the stream, captured as adults during the breeding season in 2016 for a different experiment[45]. The sex ratio was balanced in both samples.

For the release–recapture experiment, we derived from laboratory lines a large F2 hybrid population that, due to random assortment and recombination, represented a genomic mixture of lake and stream ancestry (Fig. 1). In brief, 20 lake and 20 stream individuals caught in the wild in May 2013 were crossed artificially to obtain ten unique pure-bred lake and ten stream families, from which ten F1 lake-stream hybrid families were derived in the spring of 2014 (see refs. [28,30] for details on the parental and F1 hybrid lines, and on husbandry protocols). In May 2015, 196 F2 hybrid crosses were generated based on 92 F1 hybrid females (crossed for a maximum of three times) and 58 males (siring a maximum of 6 clutches) combined haphazardly across the F1 families. The resulting F2 families were pooled haphazardly into 30 total aquaria of 55–160 L volume (5 to 15 families per aquarium). Although no effort was made to control pedigrees, our crossing protocol ensured that a reasonably high proportion of the genetic diversity of the lake and stream founder individuals was represented in the F2 hybrid population[28,30]. Laboratory mortality was negligible. All laboratory work was approved by the Veterinary Office of the Canton of Basel City.

To initiate the field experiment, all F2 hybrids were pooled in a single large oxygenated tank on 16 September 2015. From this pool, we randomly sampled 3000 individuals for the field release (Fig. 1). An additional sample of 510 individuals—our reference sample providing a baseline of the genomic composition of the F2 hybrid population before the release—was immediately killed and preserved in absolute ethanol. The following day, the release individuals were transported without mortality to the experimental site, and 1500 individuals were released above each dam (Supplementary Fig. 4). At this point, the F2 hybrids were approximately 4 months old juveniles exhibiting an average standard length of 21.9 mm (SD = 1.8; fresh body mass: 0.11 g, SD = 0.04), as measured from an additional sample (N = 30).

Recapture of the survivor F2 hybrids occurred approximately one year after the release (30 and 31 August 2016) by deploying 39 minnow traps along the full experimental stream section, including a short section immediately downstream of the lower dam. On both days, capture was performed in two rounds, each lasting c. 4 h. After each round, all stickleback captured were recorded, a small part of their anal fin was clipped and preserved individually in 100% ethanol as tissue sample, and the fish were released back into the stream. The number of new (unclipped) stickleback captured during the four rounds was 22, 12, 2 and 0; we are thus confident that sampling was (nearly) exhaustive.

**DNA library preparation and sequencing.** All our samples (natural populations, reference, and survivors) were subjected to whole-genome sequencing. For the natural populations, we generated pooled DNA libraries by first combining tissue from 10 individuals into sub-pools (24 for the lake and 23 for the stream population). To ensure a relatively similar representation among all individuals assigned to a given sub-pool, we punched a disk of 2 mm diameter from the spread caudal fin of each individual by using a biopsy plunger (KAI Medical, Chiba, Japan), and

combined these tissue samples for DNA extraction with the Qiagen DNeasy Blood & Tissue kit. We followed the manufacturer's protocol, with the modifications that the lysate resulting from proteinase digest was centrifuged and DNA was extracted from the supernatant; for the final elution, we used 60 μL of buffer for 30 min. We also included an RNAse treatment (4 μL, 100 mg/mL, for 5 min). DNA concentration was assessed with a Qubit fluorometer using the Broad Range kit (Invitrogen, Thermo Fisher Scientific, Wilmington, DE, USA), and the sub-pools were combined in equimolar proportion to two DNA libraries, one for the lake and one for the stream population. Library preparation for the F2 hybrid reference sample was performed analogously, except that we combined more individuals (15) in each of the 34 sub-pools. The latter were pooled in equimolar proportion to a single library. For the survivors, we also followed the above protocol, but to allow for greater analytical flexibility, DNA extraction was performed individually for each fin tissue sample, thus resulting in 37 separate libraries.

All DNA libraries were barcoded individually and paired-end sequenced without PCR amplification to 151 base pairs (bp) on an Illumina HiSeq2500 instrument. The two natural populations were sequenced on two lanes each, yielding an approximate read depth per base of 210× in each population, thus allowing estimating allele frequencies with high accuracy[31,46,47]. The reference library, and all survivor libraries together, were sequenced on one lane each, resulting in a mean read depth of 127x for the reference sample and 115× for all survivors combined (mean individual survivor read depth was 3.1×, range 2.6–4.4×).

**Identifying targets of selection.** We expected that if natural selection drives allele frequency shifts within a single generation, these shifts are likely small in magnitude. Hence, a plain genome-wide differentiation scan comparing the reference sample (i.e., before selection) to the survivors (after selection) was deemed unlikely to allow separating potential signatures of selection from background stochasticity. We thus considered it crucial to identify genomic regions a priori in which allele frequency shifts during our field experiment were most likely. For this, we first performed a genomic comparison of the natural populations, considering exceptionally strongly differentiated genome regions as putative targets of relatively long-term divergent natural selection between the lake and stream ecotype. Then we assessed if the reference-survivors differentiation in these regions was greater than expected by chance, which would offer evidence of selection on the released F2 hybrids.

We started by parsing all sequence output according to barcodes, followed by alignment to the third generation assembly[48] of the stickleback reference genome[12] with Novoalign 3.03.00 (Novocraft Technologies Sdn Bhd) (options: -F STDFQ -t 540 -g 40 -x 12 -r N -e 200 -i PE 200,250). Using Rsamtools[49], the resulting SAM alignments were converted to BAM format, and nucleotide counts were performed for every genomic position by applying the *pileup* function. Next, we determined the magnitude of genetic differentiation (quantified as absolute allele frequency difference AFD[31]) between the lake and stream natural populations across all genome-wide SNPs. These SNPs were required to exhibit a read depth within 100–360× in each population (thus excluding poorly sequenced and repeated regions), and a minor allele frequency (MAF) of at least 0.25 across the pool of the two populations (to ensure adequate information content[50]). This strategy yielded 1,009,247 SNPs across the 447 megabase (Mb) stickleback genome, thus resulting in one SNP per 440 bp on average.

To define candidate regions under selection, we then identified all high-differentiation SNPs in the top 0.1 percentile of the AFD distribution. This was done chromosome-specifically (autosomes only), thus taking into account variation in baseline differentiation among chromosomes due to differences in crossover rate and hence total selection density[51–53] (applying the 0.1 percentile threshold genome-wide identified a very similar set of SNPs, not presented). When high-differentiation SNPs thus identified were located within 50 kb from each other, they were treated as belonging to the same genomic region. From each independent candidate region, we finally selected the SNP exhibiting the highest differentiation plus satisfying a read depth of at least 70× and a MAF of at least 0.25 in the reference sample. In addition, this SNP was required to be sequenced in at least 35 out of the 37 survivors, thus imposing a highly stringent individual representation threshold. This yielded a panel of 126 independent high-differentiation target SNPs ascertained in the natural population comparison, at which we predicted directional selection during the experiment.

**Quantifying selection during the experiment.** To explore selection during the field experiment, we calculated for each target SNP the allele frequency shift from the reference sample to the survivors. Nucleotide counts from the individually sequenced survivors were here combined directly to a single pool, thus avoiding diploid genotype calling at low sequencing depth. The quantification of allele frequency shifts explicitly considered directionality by always expressing shifts with reference to the stream allele, that is, the allele showing a higher relative frequency in the natural stream than the lake sample.

To evaluate whether the experimental allele frequency shifts across the target SNPs were exceptional as a whole, we generated a baseline distribution for shifts at 'neutral' SNPs based on resampling. For this, we first identified all genome-wide (autosomal) SNPs falling within the AFD range of 0 to 0.1 in the natural population comparison (38% of all SNPs), assuming that these SNPs were not or little influenced by divergent selection between the lake and stream habitat. This

subset was then further restricted by retaining only those SNPs exhibiting a MAF of at least 0.35 in the reference sample. The rationale for this highly stringent MAF filtering was that the magnitude of genetic differentiation between samples is contingent on the MAF across their pool[50]: markers displaying strong differentiation necessarily also show a high MAF, whereas low differentiation is possible across a broader MAF range. Accordingly, our target SNPs—representing high-differentiation markers—showed relatively high MAFs in the reference sample (median: 0.429; Supplementary Fig. 5). With a threshold of 0.35, the MAF spectrum of our neutral SNPs closely approximated the MAF distribution observed across the target SNPs (median: 0.426; Supplementary Fig. 6). Stringent MAF filtering thus precluded that a difference in the magnitude of experimental shifts at the target versus neutral SNPs was an artifact caused by different levels of genetic diversity between these SNP classes.

From the MAF-filtered neutral SNPs, we then drew (with replacement) 9999 random samples of SNPs equal in size to the number of target SNPs (126). We here applied exactly the same standards as for the target SNPs: a physical spacing of at least 50 kb between SNPs, a minimum read depth of 70× in both the reference and survivor sample, and nucleotide counts from at least 35 survivors. Characterizing the SNP-specific reference-survivor allele frequency shifts for each of these samples finally allowed us to evaluate if the median shift observed across the target SNPs was uncommon relative to the distribution of median shifts across the neutral SNPs (throughout our study, we consider the median the most appropriate statistic of location, but additionally report the mean for key results). We emphasize that this strategy investigated a global signature of selection across the genome only; given the low expected signal-to-noise ratio, we made no attempt to infer selection on individual SNPs or genome regions.

**Robustness checks**. To assess the validity of the above statistical protocol to investigate selection during our field experiment, we implemented several alternative analyses. First, while a MAF threshold of 0.35 in the reference sample was applied to the neutral SNPs to match their MAF spectrum to the one of the target SNPs, we additionally considered MAF thresholds of 0.25 (as in the comparison of the natural populations; Supplementary Fig. 2a) and 0.4 (Supplementary Fig. 2b). The latter is extremely stringent; it raised the MAF spectrum of the neutral SNPs substantially beyond that of the target SNPs. Next, we replaced the neutral SNP panel (AFD range of 0–0.1 in the natural population comparison) by those autosomal markers deviating by no more than 25% from the genome-wide median differentiation. This corresponded to an AFD range of 0.1–0.17 (21% of all SNPs), thus producing a completely independent SNP panel for characterizing the baseline distribution of experimental allele frequency shifts (Supplementary Fig. 2c). In yet another implementation, this baseline distribution was evaluated based on SNPs drawn at random without any restriction on the magnitude of lake-stream differentiation. This latter approach was also executed with different MAF filters applied to the reference sample (0.25, Supplementary Fig. 2d; 0.35, Supplementary Fig. 2e). In all these analyses, physical spacing, read depth and survivor representation thresholds as well as the number of resampling iterations were maintained as described above. Despite the broad methodological variety covered by these robustness checks, the results remained quantitatively similar and supported identical conclusions.

**Simulations—single generation**. To develop a sense for the selection strength at individual loci required to produce a frequency shift in the order of the one observed experimentally across the target SNPs, we used individual-based forward simulations of polygenic directional viability selection over a single generation. Our base model involved a population of 1000 diploid individuals under selection at 100 independent (unlinked), biallelic, codominant loci. We chose a population size lower than the number of individuals actually released, thus taking into account the possibility that a substantial fraction of individuals may have left the experimental site in the beginning of the experiment (as there was no downstream dispersal barrier). Initial frequencies of the favorable alleles were drawn at random from the frequencies of the stream allele observed at the 126 target SNPs in the reference sample, and individual diploid multilocus genotypes were constructed according to these frequencies. Viability selection was modeled in analogy to our empirical experiment by drawing 40 individuals as survivors, the survival probability being a stochastic function of an individual's relative multilocus genotypic fitness.

Because the true link between multilocus genotype and fitness is not known for this stickleback system (or any other organism), we explored two distinct fitness functions. The first was standard multiplicative fitness, defined as $(1 − s)^n$, where $s$ is the selection coefficient and $n$ is the total number of unfavored alleles across all loci within an individual (e.g. refs. [54,55]). Key features of this fitness function are that the effect of a given allele on an individual's fitness is independent from its multilocus genetic background, and that fitness always remains positive as long as $s < 1$. The selection coefficients considered included a range of values from 0.0005 to 0.1, and were assumed to be uniform across all loci for a chosen value. The second fitness function used was additive, $1 − s*n$, with each unfavored allele reducing an individual's fitness by $s$[53,56]. With this latter fitness function, the effect of a given allele is contingent on the genetic background, thus allowing for interactions among loci. Moreover, individual fitness can here be negative. When this occurred, an individual's fitness was set to zero, thus resulting in truncation selection[43]. For additive fitness, we considered selection coefficients from 0.0005 to 0.012; with

stronger selection, the population went extinct. After viability selection, we calculated the median shift in allele frequencies in the survivors relative to the initial frequencies across all 100 loci, analogously to our empirical experiment. For each selection coefficient and fitness function, this was repeated 10,000 times, allowing plotting the grand median shift across replications against selection intensity, along with the 95% percentile band.

To check robustness, the base simulation model described above was modified by increasing the number of individuals in the beginning of the simulations from 1000 to the full release size of 3000. We also considered a higher number of loci (200) under selection, and relaxed the assumption of a uniform selection coefficient of $s$ across all loci by drawing coefficients at random from an exponential distribution with rate $1/s$. All these modifications produced results similar to the base model (Supplementary Fig. 2a–c).

**Simulations—multiple generations**. The above simulations served to explore adaptive allele frequency shifts over a single generation. An additional simulation analysis was performed to explore how rapidly locally favored alleles increase in frequency when a genetically variable population (here a genetically mixed F2 hybrid population) is exposed to polygenic directional selection associated with a novel environment (here the experimental stream habitat). As in the single-generation simulations, we assumed selection on 100 unlinked biallelic loci and considered both a multiplicative and an additive fitness scheme (simulations with 200 selected loci produced very similar results leading to the same insights; not presented). For each fitness scheme, we chose a selection coefficient $s$ plausible in the light of the single-generation simulations (Fig. 5a). That is, for multiplicative fitness, we modeled $s = 0.1$ at all loci, while $s = 0.01$ was assumed under additive fitness. For simplicity, we modeled a population of a constant size of $K = 5000$ diploid individuals ($K = 20,000$ produced very similar results, not presented). The initial allele frequencies of the locally favored alleles at all loci were specified according to the frequencies observed empirically at the 126 target SNPs in the reference population. In addition to the 100 loci under selection, we included 20 unlinked biallelic neutral loci as a control, at which initial allele frequencies were drawn from a uniform distribution bounded between 0.2 and 0.8 to ensure adequate information content.

Evolution occurred by making the probability of an individual to reproduce dependent on its multilocus genotype at the selected loci[53]. Specifically, each generation involved $K/2$ matings that produced two offspring (higher offspring numbers were modeled but produced qualitatively similar results, not presented), and individuals were recruited for mating with probabilities dependent on their relative fitness, which in turn was a direct function of the number of unfavorable alleles across their genome. Individuals were hermaphrodites and were allowed to be drawn for mating multiple times, thus causing variation among individuals in reproductive success. The transmission of alleles from parents to offspring occurred in a standard Mendelian way. With additive fitness ($s = 0.01$), a small proportion (~5%) of the population initially displayed negative fitness (which was set to zero), hence selection was truncational in the beginning.

Evolution was allowed for 500 generations. In each generation, we recorded the median frequency of the locally favored allele across all selected loci (for simplicity, we made no effort to additionally characterize statistics of dispersion in allele frequencies across loci). Allele frequency changes at the neutral loci were tracked analogously by initially defining one of the two alleles as the focal one. For each fitness scheme (and locus number), we performed 40 replicate simulations. All analyses, simulations and data visualization were performed in R version 3.6.0[57].

**Reporting summary**. Further information on research design is available in the Nature Research Reporting Summary linked to this article.

## Data availability
All raw whole-genome sequence data are available from the NCBI sequence read archive (SRA) under the accession numbers listed in the Supplementary Data 1. All input files allowing full replication of the study are provided as Supplementary Data 2 to 6.

## Code availability
All analytical code underlying this work is provided as Supplementary Software.

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

## Acknowledgements

This study was supported by Francisco Pina-Martins (coding support); Nico Boileau and Brigitte Aeschbach (wet lab support); Mirjam Bissegger (field assistance and lake stickleback sequence data generation); Hans-Peter Jermann, Roman Kistler, Milo Moser, Walter Salzburger, Markus Zellweger and Marcel Nater (logistic support, infrastructure, permits); Christian Beisel, Ina Nissen and Elodie Burcklen (Illumina sequencing at the Quantitative Genomics Facility, D-BSSE, ETH Zürich); Attila Rüegg (fish husbandry assistance); Tanja Brechbühl (sequence data archiving); Novocraft (aligner software); and Stephan Peischl (input on simulations). Computation of whole-genome data were performed at sciCORE (http://scicore.unibas.ch/), scientific computing center at University of Basel. Financial support was provided by the Burckhardt-Bürgin Foundation (T.G.L.) and the Swiss National Science Foundation (D.B.; grant 31003A_165826).

## Author contributions

Project supervision: D.B.; funding acquisition: D.B. and T.G.L.; experimental design: D.B., D.M. and T.G.L.; fish husbandry: D.M., M.A., A.F. and D.B.; field work: D.M., M.R., B.K., F.R., D.B., T.G.L. and M.A.; wet lab: T.G.L.; data analysis: T.G.L. and D.B.; visualization: T.G.L. and D.B.; coding D.B. and T.G.L.; writing: T.G.L. and D.B., with feedback from: M.R. and F.R.

## Competing interests

The authors declare no competing interests.
