## [Peer Review File · Nature Communications]

Reviewers' Comments:

Reviewer #1:

Remarks to the Author:

Laurentino et al. conducted a phylogeographic analysis of stickleback, used pooledseq to understand differences of a lake-stream natural population pair, and a release experiment of F2s from the stream-lake comparison to understand selection on stream alleles. This is exciting work that took advantage of a unique situation to conduct a natural selection experiment.

While pooled sequencing comparisons between stream and lake are appropriate for detecting allele frequency differences in natural populations, the combining of sequencing protocols, low coverage WGS for survivors, and subtle AFD between survivors and the reference population leaves much room for claiming that the release experiment resulted in allele frequency shifts, when this was due to technical or sampling variation. The shifts seen in the target SNPs average 2.3% between the reference and survivors. With such subtle shifts in allele frequency between the survivors and reference population, slight differences in pooled sequence of the reference population versus WGS of the survivors could account for the differences, especially with heterozygote drop out for the survivors given the low coverage of the WGS.

The AFD between survivors and the reference population seen in the reverse target SNPs, as demonstrated in the supplementary materials, is entirely dependent on SNP spacing threshold. They use a substantial drop in effect size when downsampling SNPs as a justification for not imposing a spacing threshold. The effect size is based on the natural allele frequency change observed in the stream-lake population comparison at the 0.1% top SNPs between the reference and the survivors relative to a resampling of all the sites available between the reference and the survivors. The reverse target and target SNPs do not overlap, thus they have little effect size to work with at these reverse target SNPs. This underscores that stochastic or technical variation could be driving these patterns, not stream-related selection. Adding to this, dispersal away from the release site was not controlled, the low-dispersal phenotypes and alleles may be enriched in the survivors.

Further in an F2 population, male and female F1s have most likely gone through 1-2 crossovers per chromosome and thus, the majority of the chromosome is still linked, and selection on one locus, will drag much of the rest of the chromosome along, giving the appearance of more genome-wide selection than is actually occurring.

L803. "Black curves"-> Dark blue?

L114. c.75m though sup Figure S1 says c. 50m - please be consistent.

L193. PCR-enrichment do you mean PCR amplification?

L360. Please report the mean and the geometric mean here as well.

L382. Please report the median and the geometric mean here as well.

L394. Why use two-tailed here? You have an apriori expectation of increased stream alleles.

L396. please also report the median and the geometric mean here as well.

L406-413. Please provide a sentence that this is entirely reliant on removing SNP spacing thresholds.

Sup material

First page of Analysis SA1: lineage->lineage

Results and conclusions: former originate->former originated

Something is wrong with this sentence "threespine stickleback are with certainty native to the

Danube"

Should mediterranean be capitalized?

It might be good to try to streamline the text of the sup material and make sentences more concise. For example, in the later part of the results and conclusions section of the phylogeography part.

Reviewer #2:

Remarks to the Author:

Review for Nature Communications

Manuscript number: NCOMMS-19-31440

Genomic release-recapture experiment in the wild reveals within-generation polygenic selection in stickleback fish

Major comments

In this study the authors conduct a manipulative experiment to understand the mode and speed of adaptation at the genomic level. A release-recapture experiment was conducted on a population of F2 hybrids of stickleback fish adapted to lake and stream habitats. The study first uses whole genome sequencing of a lake-stream population pair to identify genome-wide candidate target loci for divergent lake-stream adaptation. Next laboratory-bred F2 hybrids of the lake-stream population were exposed to a natural stream habitat for one year. Third, the study assesses variant frequency shifts in the survivors at the target loci, and after finding evidence for polygenic selection, simulations are conducted to understand the underlying selection. The authors find that adaptation in nature (shifts in adaptive allele frequency driven by selection) can be detected as a genome-wide signal over a single generation. The simulations conducted show that a large polygenic response to directional selection within a generation may be possible even with weak selection at the individual loci level.

This study addresses a key topic in evolutionary biology research – how fast natural selection influences ecologically important loci across the genome. The study incorporates a diversity of angles of investigation including utilising a comprehensive experimental setup, identification of targets of selection to test specifically, quantification of selection during the experiment, and simulations to understand the strength of selection required at individual loci to produce an average frequency shift like that observed in the experiment. I find that a thorough experimental design has been implemented, and thorough analysis of the study system, investigation of different potentially influential factors, and statistical analyses have been conducted to ensure the work is rigorous and the results are robust. For instance, because the origin of the target species in the region studied (Lake Constance Basin) is controversial, the authors first conducted molecular phylogenetic analyses (RAD markers) to understand the evolutionary relationships among stickleback fishes in Europe and also test this with more stringent parameters. Further, experimental allele frequency shifts were evaluated as well as implementing an interesting alternative analytical strategy - testing the prediction that genome regions showing the strongest experimental allele frequency shifts should also tend to show higher differentiation in the comparison of the natural populations.

Minor comments

SA1 –Using both existing and new RAD data from different studies has likely meant a relatively small number of markers (compared to what may be possible for such a system if all samples were sequenced at once currently) were used for to estimate the phylogeny. A tests of robustness is conducted using more stringent parameters and therefore fewer RAD loci. However, it would also be interesting to include comments on whether a robust dataset with a much higher number of markers would lead to the same or different phylogenetic relationship estimations.

SA3 – Simulations of polygenic evolution over multiple generations is an interesting extension to the single generation simulations presented in the manuscript, mirroring the experiment. I suggest including the multi-generation simulations in the main paper.

Overall, I found this manuscript is clear and succinct and addresses a highly relevant topic in evolutionary biology research. Care is taken to approach the research topic from different and informative angles, and to address or correct for biases or limitations in the data. I suggest this manuscript will influence the focus of future work and other research questions on this topic.

Response to reviewer comments on the Nature Communications manuscript *Genomic release-recapture experiment in the wild reveals within-generation polygenic selection in stickleback fish* by Laurentino et al.

We are thrilled to see that both reviewers and the editors share our enthusiasm about this exceptional experimental genomic study that certainly will become influential. We thank both reviewers for their thoughtful comments and valuable suggestions. We have now executed a careful revision of our paper, during which we addressed all details raised by the reviewers. This involved many phrasing improvements and additions, but the main progress is on the analytical side: we completely re-implemented our core pipeline performing the test of selection in action by adding further quality control and several robustness checks. We are confident that we now do the best possible job in excluding any bias that could influence our result. To our satisfaction, the evidence of selection over a single generation has only become stronger with these methodological improvements. We again thank the reviewers for motivating all improvements, and hope that our work is now ready for acceptance.

Below, we provide case-by-case responses to the reviewer comments. Their comments are printed in blue in the original wording, and our responses are printed black. Line numbers refer to the revised manuscript.

Reviewer 1

Laurentino et al. conducted a phylogeographic analysis of stickleback, used pooledseq to understand differences of a lake-stream natural population pair, and a release experiment of F2s from the stream-lake comparison to understand selection on stream alleles. This is exciting work that took advantage of a unique situation to conduct a natural selection experiment.

While pooled sequencing comparisons between stream and lake are appropriate for detecting allele frequency differences in natural populations, the combining of sequencing protocols, low coverage WGS for survivors, and subtle AFD between survivors and the reference population leaves much room for claiming that the release experiment resulted in allele frequency shifts, when this was due to technical or sampling variation. The shifts seen in the target SNPs average 2.3% between the reference and survivors. With such subtle shifts in allele frequency between the survivors and reference population, slight differences in pooled sequence of the reference population versus WGS of the survivors could account for the differences, especially with heterozygote drop out for the survivors given the low coverage of the WGS.

The major concern of Reviewer1 (R1) is the combination of poolSeq (for the reference sample) and individual-level sequencing ('indSeq', for the survivors) for investigating selection during the experiment. In particular, diploid genotype calling based on indSeq at low read depth may cause heterozygote dropout. This may have produced a spurious allele frequency shift at the selection targets in our study. This view has several flaws:

First, even if there is heterozygote dropout, this dropout must be random with respect to genotype within an individual. Hence, heterozygote dropout could in theory generate noise in allele frequency estimation across a sample of individuals, but not bias these frequencies.

Second, there cannot be heterozygote dropout in our study for the simple reason that we never called individual-level diploid genotypes across our analytical pipeline. The full raw survivor allele counts were combined without any manipulation to a survivor pool, thus leading to

a data set absolutely comparable to the reference pool. We recognize, however, that this point was not explicit enough in the previous manuscript, hence the revised version now contains an unambiguous statement that we pooled raw base counts for the survivors (L 399-402). The reason why we performed indSeq for the survivors was just to be able to control the number of individuals contributing sequence data to the survivor pool (stated on L 342-345), and we performed this control highly stringently (base counts from at least 35 out of 37 individuals required; L 392 and 429). We also note that read depth was highly consistent among individuals (declared on L 350-353), so that we lost few markers while filtering for survivor representation.

The third challenge to R1's concern is that the key evidence in our analysis stems from the comparison of target SNPs versus baseline SNPs. So even if there was a hypothetical problem with the genotyping of the survivors, this would affect both classes of SNPs, and hence would not explain why target SNPs on average show a shift in favor of the stream allele.

Finally, we highlight that although an average shift in the order of 2.5% may seem subtle, one should keep in mind that we avoid single-locus inference. The analytical power of our study comes from our simultaneous focus on 126 target SNPs at which selection was predicted *a priori*. Moreover, although data from 35-37 survivors at a minimum of 70x read depth is modest compared to our natural population comparison (>220 individuals per population, >210x read depth per population), we highlight that genome scans performed with such quality are rare. According to poolSeq theory, our survivor data are certainly expected to allow reasonably precise allele frequency estimates (some key references on this point are given on L 350 references: 31, 46 and 47).

The above arguments and improvements to the manuscript should resolve R1's concern. That said, while the manuscript was in review, we realized that there was one potential source of bias in our analysis that had been overlooked. The issue is the minor allele frequency (MAF) spectrum of the two classes of SNPs compared in the experiment: the target SNPs are by definition markers showing high differentiation between the natural populations. Consequently, they must all exhibit a relatively high MAF across the population pool. By contrast, selecting loci at random across the genome for estimating the baseline shift will mostly pick SNPs showing weak population differentiation, and these may show a MAF spectrum systematically shifted toward lower values. Because the MAF spectrum can influence the magnitude of population differentiation (for details see our well-cited methodology paper; Roesti et al. 2012, BMC Evol. Biol.), we considered it possible that the shift at the target SNPs was an artifact of different MAF spectra between target and baseline markers.

To evaluate this concern, we completely re-implemented the evaluation of the target SNPs against the baseline distribution, this time controlling the MAF spectrum of the baseline SNPs (L146-149; Figure 4 and Figure S4), and additionally performed extensive robustness checks that now form a separate section in the Methods ('**Robustness checks**'; L438-458) and an item in the Supporting Information. To our relief, all these new analyses only confirmed the robustness of our initial conclusion, irrespective of the MAF spectrum of the resampled SNPs: the overall shift in favor of the stream alleles observed at our target SNPs is highly unlikely to arise from chance. We note that due to a minor algorithm improvement (change of the order in which quality filters are applied), the new analysis is based on a higher number of target SNPs (increase from 78 to 126), which also makes our analysis more sensitive (hence leading to an even lower probability of observing our pattern by chance; $P = 0.0173$, L 146-149).

By resolving the MAF issue through extensive re-implementation and -analysis, we are confident that every possible source of bias that may have produced an artifact is now ruled out. We really see no way in which the quality of the evidence could be raised further. We can thus assure R1 that his/her concern is unfounded.

The AFD between survivors and the reference population seen in the reverse target SNPs, as demonstrated in the supplementary materials, is entirely dependent on SNP spacing threshold. They use a substantial drop in effect size when downsampling SNPs as a justification for not imposing a spacing threshold. The effect size is based on the natural allele frequency change observed in the stream-lake population comparison at the 0.1% top SNPs between the reference and the survivors relative to a resampling of all the sites available between the reference and the survivors. The reverse target and target SNPs do not overlap, thus they have little effect size to work with at these reverse target SNPs. This underscores that stochastic or technical variation could be driving these patterns, not stream-related selection.

We think the main issue with the reverse target SNP analysis is that this alternative SNP panel contains just a lot of stochastic noise. Hence our initial idea to maximize sample size by not applying any physical spacing threshold. We agree with R1, however, that probably a spacing threshold would have been needed to reach independence among SNPs, as in the main target SNP analysis. We have thus redone the reverse target SNP evaluation by enforcing a spacing threshold (our study-wide new standard is 50 kb). However, doing so reduced our SNP number to 286. Given the subtlety of the overall selective allele frequency shift during our experiment (2.5 percent), it is clear that with 37 individuals, a SNP panel of that size must be dominated by stochastic noise. (Note that the target SNP panel was chosen based on allele frequency data from 229 and 240 individuals per population sequenced to 210x, and based on the signature of long-term selection, thus leading to excellent precision in identifying SNPs under selection.) For this reason, we have now decided to omit the reverse target SNP analysis. Our re-analysis still recovered elevated differentiation in the natural population scan at the reverse target SNPs by 6% over the background (in our previous analysis, this was 8%), but statistically, it appeared difficult to make a sound judgment about this quantity. Given the high quality and stringency of our main analysis (target SNPs), we feel that the reverse target SNP analysis is not needed and that its exclusion is no loss.

Adding to this, dispersal away from the release site was not controlled, the low-dispersal phenotypes and alleles may be enriched in the survivors.

Yes, indeed genotype-dependent dispersal may well have contributed to the enrichment of stream alleles observed in our experiment. R1 may have overlooked, however, that we address this possibility explicitly in a paragraph of the Discussion (L 223-237). In that passage, we explain why such dispersal must be understood as a form of natural selection; we intentionally allowed for genotype-dependent dispersal in our experiment because this is an interesting potential component of adaptation. The possibility of such dispersal does not weaken or challenge our investigation in any way.

Further in an F2 population, male and female F1s have most likely gone through 1-2 crossovers per chromosome and thus, the majority of the chromosome is still linked, and selection on one

locus, will drag much of the rest of the chromosome along, giving the appearance of more genome-wide selection than is actually occurring.

We appreciate this reflection, but there is a crucial detail that R1 may have overlooked and that settles this concern: the natural lake and the stream population underlying our F2 hybrid population are far from being completely differentiated at the loci under divergent selection. Specifically, these latter loci – probed by our 126 target SNPs assessed for evolution during the field experiment, displayed a median (and average) allele frequency differentiation between the natural lake and stream population of ‘only’ 0.139 and 0.165, respectively (stated on L 90-91). This implies that lake- and stream-favored alleles must still be segregating and recombining to some extent *within* each habitat. In other words, the haplotype-level diversity needed for a polygenic response to selection, as captured by our experiment, was already pre-existing in these natural populations. Since our F2 population was founded from 10 replicate unique F1 lake-stream hybrid families, a rich haplotype diversity in our F2 hybrids can be taken for granted. The main benefit of our production of F2 hybrids was therefore not to produce haplotype diversity by linking lake and stream chromosome segments (although we certainly did a little bit of that), but simply to drive the overall frequency of stream-adapted alleles away from their frequency in the natural population (which we did very effectively; L 131-137, Fig. 3a).

R1’s comment made us aware that this aspect is perhaps not evident without a specific note. We have therefore added a passage declaring that ample haplotype diversity must have been present in the founders of our laboratory lines already, and was not primarily generated during our laboratory crossing (L 137-142).

L803. "Black curves"-> Dark blue?

Thanks for spotting – is corrected for all figures.

L114. c.75m though sup Figure S1 says c. 50m - please be consistent.

Thanks for pointing out, we have made consistent (L 272).

L193. PCR-enrichment do you mean PCR amplification?

Yes, amplification. We agree with R1 that amplification is more precise, hence have re-phrased (L 347).

L360. Please report the mean and the geometric mean here as well.

We appreciate that R1 raises the issue of the statistic of location. Indeed, our previous manuscript was slightly inconsistent in that we sometimes presented (arithmetic) means, sometimes medians. However, based on a fair bit of experience in genomic data analysis, we argue that the median is generally the most meaningful statistic of location, as it is more robust to tailed distributions than the mean. For our revision, we have therefore re-implemented *all* analyses with the median as statistic of location. However, we have also re-done all analyses with the arithmetic mean, and this does not change any result. Our key test of selection in the wild, for instance, yields a P value of 0.017 with the median, and 0.006 with the arithmetic mean, thus leading to the same conclusion with both statistics. Given that R1 asks for the presentation of mean values too, we have added the arithmetic mean in addition to the median for the key results (L 91 on natural population differentiation, L 111 on target SNP AFD, L 146 on the magnitude of the observed frequency shift and L 149 on the two-tailed probability of observing

the empirical shift). And to avoid confusion, we have added to the Methods a general declaration that the median is our statistic of choice, but that we also report the mean for key results (L 429-434).

As to R1's recommendation to additionally report geometric means, we find this suggestion confusing for several reasons: First, we are unaware of any empirical genomic investigation in which the geometric mean was used as a statistic of location. Actually, we had to search the typical applications and definition of the geometric mean ourselves, and we suspect that many readers also would not have a clear idea about what the geometric mean is, or how it is calculated. Second, the geometric mean includes the product of the values of a data vector, thus leading to enormously large numbers if one works with hundred thousands of data points. Consequently, we simply did not manage to calculate the geometric mean for several of our relevant larger data sets (e.g., genome-wide mean differentiation between the natural populations) for computational reasons. Third, the geometric mean is undefined for negative values, thus precluding its application to some of our key data sets (most notably the experimental shifts in allele frequencies, which can be positive or negative). Finally, for our smallest data set, the 126 target SNPs, we did manage to compute the geometric mean for the magnitude of differentiation in the natural populations, and obtained a value almost identical to our median and arithmetic mean: median = 0.589, arithmetic mean = 0.602, geometric mean = 0.597.

Given these considerations, we do not understand why R1 suggests using the geometric mean; they do not indicate the reason why this highly uncommon statistic would be valuable. We therefore prefer presenting our work based on the median as our main location statistic, sometimes complemented by the arithmetic mean.

L382. Please report the median and the geometric mean here as well.

See our response to the previous point.

L394. Why use two-tailed here? You have an a priori expectation of increased stream alleles.

R1's suggestion to use a one-tailed evaluation of the null hypothesis is problematic; it can be considered a manipulation of a statistical test such that the resulting P value drops below, or is further away from, some significance threshold (that is, P-hacking). The objective of such manipulations is to make a scientific result appear more certain than it is, at the risk of selling a false positive.

The problem with R1's suggestion to use one-tailed evaluation is that in almost every biological investigation, researchers could claim to have a strong 'a priori expectation' about the direction of an effect. Such expectations are typically the very motivation to conduct research in the first place. However, biological research is generally confronted with noise and unknown confounding variables, so that one-tailed testing is – rightly – discredited in our field (for details on the inappropriate use and inadequate justification of one-tailed tests see e.g. Lombardi and Hulbert, 2009 <https://onlinelibrary.wiley.com/doi/full/10.1111/j.1442-9993.2009.01946.x>).

In brief, one-tailed testing produces overconfidence in results, increases the risk of false discoveries, and is therefore poor practice. In our analytical context, there is no physical impediment for allele frequencies to shift towards lake-like frequencies instead of stream-like. Hence, despite our expectation that stream alleles would increase in frequency, it is imperative

for the evaluation of the observed pattern that the two directions of allele frequency change are considered. We thus maintain two-tailed evaluation throughout the manuscript.

L406-413. Please provide a sentence that this is entirely reliant on removing SNP spacing thresholds.

This comment refers to the reverse target SNP analysis that has now been excluded from the study (please see our second response to R1 above).

Sup material

First page of Analysis SA1: lineage→lineage

Results and conclusions: former originate->former originated

Something is wrong with this sentence "threespine stickleback are with certainty native to the Danube"

Should mediterranean be capitalized?

It might be good to try to streamline the text of the sup material and make sentences more concise. For example, in the later part of the results and conclusions section of the phylogeography part.

All of these comments refer to the phylogeographic analysis exploring the historical background of our study populations, formerly presented within the Supporting Information. However, this analysis has now been removed from the paper; it will form a main component of a separate publication. There are strong reasons for this decision: several authors of the present study found that this analysis was somewhat off topic and distracting. Indeed, the history of the study populations has absolutely no bearing on the study design, the methods, analyses, nor the interpretation of the results (outside the Methods, the previous manuscript never referred to the phylogeographic analysis).

The motivation for this phylogeographic analysis was to address previous confusion and controversy around the stickleback of the Lake Constance basin. Overall, we considered the present study the best – yet not ideal – venue for that investigation. However, in the meantime, we have decided to write a manuscript exclusively focused on the phylogeography of stickleback in the Constance basin in response to a new paper that, in our view, presents misleading interpretations on the topic. Hence it is no longer necessary to include the phylogeographic analysis in the present paper. We hope that the reviewers and editors agree that this decision increases the clarity and conciseness of our present selection-in-action study.

Reviewer 2

Major comments

In this study the authors conduct a manipulative experiment to understand the mode and speed of adaptation at the genomic level. A release-recapture experiment was conducted on a population of F2 hybrids of stickleback fish adapted to lake and stream habitats. The study first uses whole genome sequencing of a lake-stream population pair to identify genome-wide candidate target loci for divergent lake-stream adaptation. Next laboratory-bred F2 hybrids of the lake-stream population were exposed to a natural stream habitat for one year. Third, the study assesses variant frequency shifts in the survivors at the target loci, and after finding evidence for polygenic selection, simulations are conducted to understand the underlying selection. The

authors find that adaptation in nature (shifts in adaptive allele frequency driven by selection) can be detected as a genome-wide signal over a single generation. The simulations conducted show that a large polygenic response to directional selection within a generation may be possible even with weak selection at the individual loci level.

This study addresses a key topic in evolutionary biology research – how fast natural selection influences ecologically important loci across the genome. The study incorporates a diversity of angles of investigation including utilising a comprehensive experimental setup, identification of targets of selection to test specifically, quantification of selection during the experiment, and simulations to understand the strength of selection required at individual loci to produce an average frequency shift like that observed in the experiment. I find that a thorough experimental design has been implemented, and thorough analysis of the study system, investigation of different potentially influential factors, and statistical analyses have been conducted to ensure the work is rigorous and the results are robust. For instance, because the origin of the target species in the region studied (Lake Constance Basin) is controversial, the authors first conducted molecular phylogenetic analyses (RAD markers) to understand the evolutionary relationships among stickleback fishes in Europe and also test this with more stringent parameters. Further, experimental allele frequency shifts were evaluated as well as implementing an interesting alternative analytical strategy - testing the prediction that genome regions showing the strongest experimental allele frequency shifts should also tend to show higher differentiation in the comparison of the natural populations.

Minor comments

SA1 –Using both existing and new RAD data from different studies has likely meant a relatively small number of markers (compared to what may be possible for such a system if all samples were sequenced at once currently) were used for to estimate the phylogeny. A tests of robustness is conducted using more stringent parameters and therefore fewer RAD loci. However, it would also be interesting to include comments on whether a robust dataset with a much higher number of markers would lead to the same or different phylogenetic relationship estimations.

There is a recent world-wide phylogeographic study (Fang et al. 2018) that used a massively higher marker number than our phylogeographic work. We used a fair subset of individuals from that study, but intersecting these data with additional data sets from other studies caused a large marker loss. Nevertheless, the tree topology of our analysis is fully consistent with the one recovered in Fang et al. (2018). We are thus confident that our phylogeographic inference is robust. That said, we note that the phylogeographic analysis is no longer part of our selection study (please see our last response to R1 for details).

SA3 – Simulations of polygenic evolution over multiple generations is an interesting extension to the single generation simulations presented in the manuscript, mirroring the experiment. I suggest including the multi-generation simulations in the main paper.

Great suggestion, appreciated. We agree that this simulation conveys an important finding: based on selection coefficients realistic given our empirical work, a population can adapt very rapidly genome-wide to a new environment. Since strong phenotypic evolution in stickleback has

indeed been observed on the time scale of dozens of generations, this multigenerational simulation analysis provides an additional link between genomic and phenotypic evolution. We have followed R2's recommendation to include this analysis in the main paper (in Methods L 504 - 542; in Results L 181-200; Figure 5b).

Overall, I found this manuscript is clear and succinct and addresses a highly relevant topic in evolutionary biology research. Care is taken to approach the research topic from different and informative angles, and to address or correct for biases or limitations in the data. I suggest this manuscript will influence the focus of future work and other research questions on this topic.